# Analysis of the Behavioral Change and Utility Features of Electronic Activity Monitors

**Zakkoyya H. Lewis** [1,*], **Maddison Cannon** [1], **Grace Rubio** [1], **Maria C. Swartz** [2] **and Elizabeth J. Lyons** [3]

[1] Department of Kinesiology and Health Promotion, College of Science, California State Polytechnic University Pomona, 3801 West Temple Ave., Pomona, CA 91768, USA; Madkcan20@gmail.com (M.C.); graciekhp17@gmail.com (G.R.)

[2] Department of Pediatrics, Division of Pediatrics, MD Anderson Cancer Center, 7777 Knight Rd., Houston, TX 77054, USA; MChang1@mdanderson.org

[3] Department of Nutrition and Metabolism, School of Health Professions, University of Texas Medical Branch, 301 University Blvd., Galveston, TX 77555, USA; ellyons@utmb.edu

\* Correspondence: zakkoyyal@cpp.edu

**Abstract:** The aim of this study was to perform a content analysis of electronic activity monitors that also evaluates utility features, code behavior change techniques included in the monitoring systems, and align the results with intervention functions of the Behaviour Change Wheel program planning model to facilitate informed device selection. Devices were coded for the implemented behavior change techniques and device features. Three trained coders each wore a monitor for at least 1 week from December 2019–April 2020. Apple Watch Nike, Fitbit Versa 2, Fitbit Charge 3, Fitbit Ionic—Adidas Edition, Garmin Vivomove HR, Garmin Vivosmart 4, Amazfit Bip, Galaxy Watch Active, and Withings Steel HR were reviewed. The monitors all paired with a phone/tablet, tracked exercise sessions, and were wrist-worn. On average, the monitors implemented 27 behavior change techniques each. Fitbit devices implemented the most behavior change techniques, including techniques related to the intervention functions: education, enablement, environmental restructuring, coercion, incentivization, modeling, and persuasion. Garmin devices implemented the second highest number of behavior change techniques, including techniques related to enablement, environmental restructuring, and training. Researchers can use these results to guide selection of electronic activity monitors based on their research needs.

**Keywords:** activity tracker; wearable; physical activity; behavior change technique

## 1. Introduction

The benefits to health and overall well-being from regular physical activity (PA) are well established [1,2]. However, physical inactivity is on the rise and is the fourth leading cause of global mortality [2]. Researchers have conducted behavioral intervention studies for decades in an attempt to increase PA. These studies have often recruited the assistance of mobile technologies such as mobile phones, websites, and e-mails to deliver their PA interventions [3,4]. One commonly used form of technology is the electronic activity monitor (EAM) [5,6]. EAMs have also been referred to as "lifestyle activity monitors", "activity tracker" or "wearable". The key feature of an EAM versus other activity trackers (e.g., pedometers) is that it meets the following definition: objectively measured lifestyle PA and can provide feedback, beyond the display of basic activity count information, via the monitor display or through a partnering application (app) to elicit continual self-monitoring of activity behavior [5]. EAMs have been shown to have great potential as an adjuvant tool to increase PA in

behavioral interventions [5–8]. Conventional PA interventions often rely on extensive behavioral counseling, which can be time-consuming for the interventionist [9]; the addition of an EAM allows for continued behavior modification while alleviating the time burden of an interventionist [10]. EAMs also allow for the wearer's behavior to be more accessible to them through self-monitoring and allow for real-time feedback, which may promote autonomy. Results from a recent meta-analysis suggest that EAM use may result in an increase of 40 min of moderate-to-vigorous PA per day (standardized mean difference = 0.17) or up to 92 min per day (standardized mean difference = 0.33) when used as a component of a multifaceted intervention [7]. The success of EAMs within interventions may be due in part to the embedded behavior change techniques (BCTs), which are known to promote PA [11,12].

A taxonomy of BCTs was developed by Michie and colleagues after systematic investigation of behavior change interventions. The results of their meta-regression identified effective intervention components or techniques that promote physical activity [13]. The taxonomy includes 93 well-defined BCTs across 16 groups [14]. BCTs can be implemented through traditional behavioral interventions or through behavioral devices such as EAMs. Lyons et al. first published a content analysis that reviewed the BCTs embedded in well-known EAMs in 2014 [12]. At that time, EAMs were starting to proliferate commercially and in research. Other researchers have continued to investigate implemented BCTs in EAMs [15–17] but it is difficult for research to keep up with advances in technology. Most devices that were included in the early publications are either no longer available or are now obsolete. Furthermore, the analyses are limited in their scope. The results were presented without context of how a given BCT can enhance an intervention. Selecting an EAM with the appropriate BCTs requires informed decision making that is dependent on the research design and the preferences of the wearer.

## 1.1. Informed Decision Making

The success of a PA behavior change intervention is dependent on a robust intervention design. When designing studies, there is a need to appropriately characterize intervention components and link them to the targeted PA behavior [18]. Several frameworks are available to classify behavior change interventions but one system in particular offers clear connections between theoretical constructs, intervention strategies, and specific BCTs for implementing those strategies—the Behaviour Change Wheel (BCW) [18]. The BCW was developed after the systematic evaluation of other behavior change frameworks and aimed to address their limitations [18]. At its core, the BCW is based on the integrative model: Capability Opportunity Motivation—Behaviour (COM-B). The COM-B constructs correspond with intervention functions that are directly linked to BCTs [18,19]. These intervention functions include increasing knowledge (education), communicating to induce feelings or stimulate action (persuasion), creating expectation of reward (incentivization), creating expectation of punishment or cost (coercion), imparting skill (training), reducing barriers to increase opportunity (enablement), providing an example to imitate (modeling), and changing the physical/social environment (environmental restructuring) [18,19]. BCTs are intervention strategies that target the aforementioned intervention functions. The BCW matches the target behavior, intervention function, and the desired BCTs [19]. For example, if a researcher or health practitioner determines a participant would benefit from more education, they would identify BCTs such as information on health consequences and prompts/cues. Alternatively, if a participant needs assistance in reducing barriers then the researcher or health practitioner can identify enablement BCTs such as goal setting and graded tasks. By intentionally selecting an intervention function within the BCW, researchers should also be able to choose an appropriate EAM based on the embedded BCTs within the device and/or its associated application (app).

The success of EAMs is also determined by factors that may impact wearers' engagement [10]. Factors that can impact wearers' engagement include, but are not limited to, measurement validity, social functionality, aesthetics, the physical form of the device, feedback, readability, and gamification [20,21]. Gamification is executed, in part, through the implemented BCTs whereas the other factors are based on practicality and utility. A survey of EAM users identified several practical features that impact how wearers use their device; these features include functionality (e.g., battery life, wear location, device

pairings) and monitored behaviors [22]. Before researchers can determine whether there is a correlation between utility features and wearers' engagement, there is a need to catalog the utility features present in EAMs. To our knowledge, there is no systematic evaluation of these EAM utility features.

### 1.2. Study Aim

The aim of the current study was to perform an updated behavioral content analysis of EAMs currently on the market. Our analysis was expanded to include a systematic review of device utility features and the results are aligned with intervention functions within the BCW. This was completed to help align EAM features with intervention needs. Researchers and health practitioners can use the results to make an informed selection of an EAM for physical activity promotion.

## 2. Methods

EAMs were identified using the CNet list of "Best Wearable Tech for 2020" and the associated buying guide [23]. This strategy for identifying EAMs is common practice for this type of research [12,16]. The list included several models from the same manufacturer (e.g., Fitbit Ionic, Fitbit Versa) and different versions of the same device (e.g., Apple Watch Series 4, Apple Watch Series 3). Only the latest version of the device was included in the current review to evaluate the latest features. Different models from the same manufacturer were included because of the difference in utility features. EAMs included in this review were Apple Watch Nike Series 5, Fitbit Versa 2, Fitbit Charge 3, Fitbit Ionic—Adidas Edition, Garmin Vivomove HR, Garmin Vivosmart 4, Amazfit Bip, Galaxy Watch Active, and Withings Steel HR. The manufacturer and compatibility information for each device is listed in Table 1.

BCTs were coded based on the behavior change taxonomy created by Michie et al. (2013) and they were further separated by the intervention functions according to the BCW [19]. Utility features were coded based on a list of features reported in a survey of EAM users [22]. Coding was based on whether or not the BCT or feature was present. Three trained coders (ZHL, MC, GR) each wore a monitor for at least 1 week from December 2019–April 2020. The coders included the Principal Investigator, a PhD-level researcher, and two senior-standing undergraduate research assistants that completed coursework in exercise behavior. All coders had experience using EAMs and were well-versed on BCT definitions as well as examples of how they were embedded in EAMs. Each device was coded by two blinded coders and reviewed by a third coder who was unblinded to the results of the previous two coders. The addition of the third coder allowed for (1) capturing any BCT or feature that was present but was not recorded by the other coders and (2) settling any discrepancies between coders. If a BCT was identified by at least two reviewers, it was coded as present. Inter-coder reliability was determined using the kappa statistic for codes identified by at least two reviewers versus codes identified by one reviewer [24]. ZHL was a blind coder for all devices while MC and GR alternated between the blinded and unblinded coder for a given device (see Additional File S1 for coding schedule).

Coders downloaded partnering apps on their personal mobile device for each monitor. This allowed for a review using two different operating systems, iOS (ZHL) and Android Operating System (MC and GR). Codes were the same for devices from the same manufacturer (e.g., Fitbit, Garmin) as the BCTs are primarily delivered through the app. When additional payment was required to access content (e.g., Fitbit Premium features), codes were based on the free features available. This allowed for a complete list of the minimum available BCTs for each manufacturer. Where functionality existed but was not necessarily a default feature, it was coded as present. For example, "friends" are available for social support, but the user must add friends and "insights" are available to provide detailed feedback, but the user must enroll in this feature. Additional File S2 provides a complete coding for each EAM with codes aggregated from all coders.

**Table 1.** Electronic activity monitor (EAM) manufacture and compatibility information.

| EAM | Manufacturer | Location | Partnering App | Compatible Operating System | Unit Price (USD) * |
|---|---|---|---|---|---|
| Amazfit Bip | Huami | Hefei, China | Amazfit | iOS 8+<br>Android OS 4.4+ | $79.99 |
| Apple watch Nike, series 5 | Apple | Cupertino, CA USA | Watch | iOS 13+<br>(only with iPhone 6 or greater) | $429.00 |
| Fitbit Charge 3<br>Fitbit Ionic—Adidas<br>Fitbit Versa 2 | Fitbit | San Francisco, CA USA | Fitbit | iOS 12.2+ Android OS 7.0+<br>Windows 10v1607+ | $149.95 (Charge 3)<br>$279.95 (Ionic)<br>$199.95 (Versa 2) |
| Galaxy Watch Active | Samsung | Seoul, South Korea | Galaxy Watch | iOS 9.0+<br>Android OS 5.0+ | $199.99 |
| Garmin Vivomove HR<br><br>Garmin Vivosmart 4 | Garmin | Olathe, KS USA | Garmin Connect | iOS 12+<br>Android OS 5.0+ | $349.99 (Vivomove HR)<br><br>$129.99 (Vivosmart 4) |
| Withings Steel HR | Withings | Issy-les-Moulineaux, France | Health Mate | iOS 10+<br>Android OS 6.0+ | $224.99 |

* at the time of purchase in December 2019.

It is important to note that coding was incomplete for the Apple watch and the Galaxy watch. The Apple watch was not coded by multiple reviewers due to compatibility issues. The device is only functional once paired with an iPhone 6 or greater. The Galaxy watch has limited functionality when paired with iOS products. Coding was still completed by all three coders with all available features accessible to two coders (MC, GR).

## 3. Results

Inter-coder reliability ranged from slight to almost perfect (Amazfit Bip, $\kappa = 0.35$; Fitbit, $\kappa = 0.16$; Galaxy Watch, $\kappa = 0.37$; Garmin, $\kappa = 0.96$; Withings Steel HR, $\kappa = 0.50$). Table 4 displays the BCTs implemented in each EAM based on the corresponding intervention function defined by the BCW. The complete list of implemented BCTs is available in Additional File S1. On average, 27.7 BCTs were implemented across all EAM apps. Examples of how the BCTs were implemented for each device are available in Additional File S3. All devices included the following BCTs: goal setting (behavior), review behavior goal(s), discrepancy between current behavior and goal, feedback on behavior, self-monitoring of behavior, biofeedback, social support (unspecified), social comparison, prompts/cues, non-specific reward, restructuring the physical environment, and adding objects to the environment. Overall, the Fitbit devices implemented the most BCTs ($n = 43$), followed by Garmin ($n = 36$), Amazfit Bip ($n = 25$), Galaxy Watch ($n = 23$), Apple Watch ($n = 20$), and Withings Steel HR ($n = 19$). Figure 1 displays the number of BCTs implemented in each device by BCW intervention function. The total number of BCTs depicted in the figure accounts for some BCTs appearing in multiple intervention functions. Each intervention function has an allotted number of possible associated BCTs: education (15 BCTs), enablement (46 BCTs), environmental restructuring (11 BCTs), coercion (18 BCTs), incentivization (30 BCTs), modeling (1 BCT), persuasion (16 BCTs) and training (15 BCTs) [19]. No device implemented all associated BCTs for a given intervention function. Fitbit, through its device and the associated app, implemented the most BCTs that support the BCW intervention functions of education, enablement, environmental restructuring, coercion, incentivization, modeling and persuasion. Garmin implemented the most BCTs that support enablement, environmental restructuring, and training.

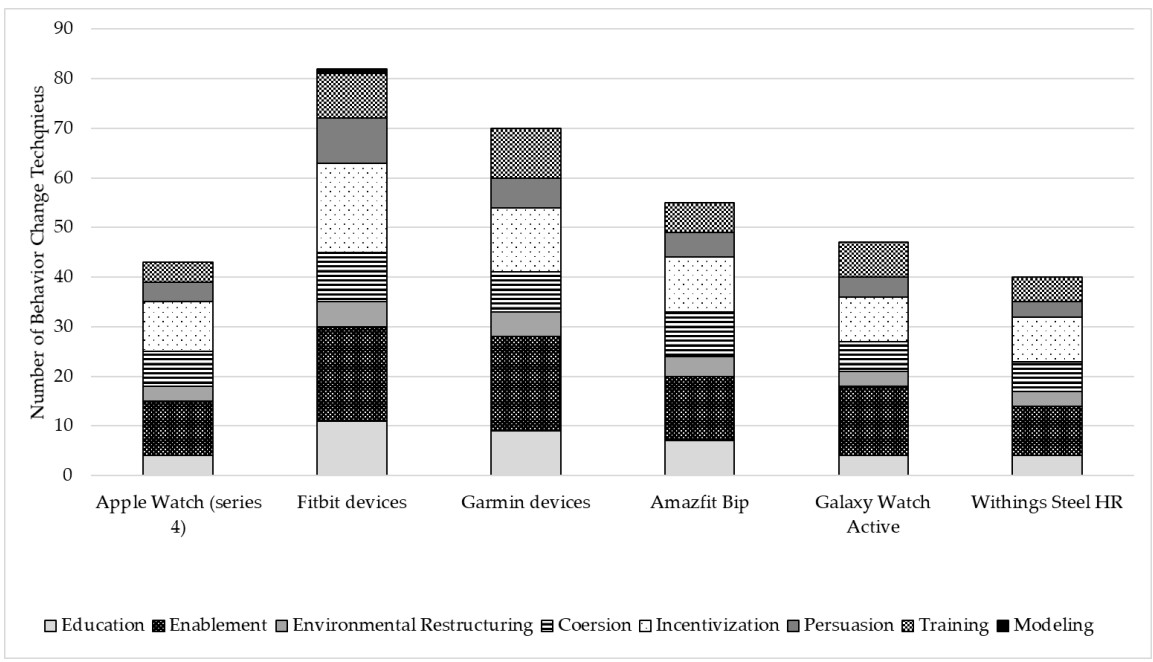

**Figure 1.** Frequency of behavior change techniques by intervention function within EAMs.

**Table 2.** EAM coding based on behavior change techniques (BCTs) associated with intervention functions of the Behaviour Change Wheel (BCW).

| Behavior Change Technique | Apple Watch (Series 4) | Fitbit Devices | Garmin Devices | Amazfit Bip | Galaxy Watch Active | Withings Steel HR | Total |
|---|---|---|---|---|---|---|---|
| **EDUCATION: increasing knowledge *** | | | | | | | |
| 2.2 Feedback on behavior | X | X | X | X | X | X | 6 |
| 2.3 Self-monitoring of behavior | X | X | X | X | X | X | 6 |
| 2.4 Self-monitoring of outcome(s) of behavior | | X | X | X | | | 3 |
| 2.6 Biofeedback | X | X | X | X | X | X | 6 |
| 2.7 Feedback on outcome(s) of behavior | | X | | X | | | 2 |
| 4.2 Information about antecedents | | X | X | | | | 2 |
| 4.4 Behavioral experiments | | | X | | | | 1 |
| 5.1 Information about health consequences | | X | X | | | | 2 |
| 5.6 Information about emotional consequences | | X | | | | | 1 |
| 6.3 Information about others' approval | | X | | | | | 1 |
| 7.1 Prompts/cues | X | X | X | X | X | X | 6 |
| 7.2 Cue signaling reward | | X | X | X | | | 3 |
| Total | 4 | 11 | 9 | 7 | 4 | 4 | |
| **ENABLEMENT: reducing barriers *** | | | | | | | |
| 1.1 Goal setting (behavior) | X | X | X | X | X | X | 6 |
| 1.3 Goal setting (outcome) | | X | X | X | X | | 4 |
| 1.5 Review behavior goal(s) | X | X | X | X | X | X | 6 |
| 1.6 Discrepancy between current behavior and goal | X | X | X | X | X | X | 6 |
| 1.7 Review outcome goal(s) | | X | | X | X | | 3 |
| 1.8 Behavioral contract | X | X | | X | X | X | 5 |
| 1.9 Commitment | X | X | X | | X | X | 5 |
| 2.3 Self-monitoring of behavior | X | X | X | X | X | X | 6 |
| 2.4 Self-monitoring of outcome(s) of behavior | | X | X | X | | | 3 |
| 3.1. Social support (unspecified) | X | X | X | X | X | X | 6 |
| 3.3. Social support (emotional) | | X | X | X | | | 3 |
| 4.4 Behavioral experiments | | | X | | | | 1 |
| 5.5 Anticipated regret | | X | | | | | 1 |
| 8.2 Behavior substitution | | X | | | X | | 2 |
| 8.7 Graded tasks | X | X | X | | | | 3 |
| 9.3 Comparative imagining of future outcomes | | | X | | | | 1 |
| 10.9 Self-reward | | | | | X | | 1 |
| 11.2 Reduce negative emotions | | - | X | | | X | 2 |
| 11.3 Conserving mental resources | | X | X | | | | 2 |
| 12.1 Restructuring the physical environment | X | X | X | X | X | X | 6 |
| 12.2 Restructuring the social environment | | X | | | | | 1 |
| 12.5 Adding objects to the environment | X | X | X | X | X | X | 6 |
| 13.2 Framing/reframing | | | X | | | | 1 |
| 15.2 Mental rehearsal of successful performance behavior | | | X | | | | 1 |
| 15.3 Focus on past success | X | X | X | X | X | | 5 |
| Total | 11 | 19 | 19 | 13 | 14 | 10 | |

**Table 3.** EAM coding based on behavior change techniques (BCTs) associated with intervention functions of the Behaviour Change Wheel (BCW).

| Behavior Change Technique | Apple Watch (Series 4) | Fitbit Devices | Garmin Devices | Amazfit Bip | Galaxy Watch Active | Withings Steel HR | Total |
|---|---|---|---|---|---|---|---|
| **ENVIRONMENTAL RESTRUCTURING: Change the physical or social environment \*** | | | | | | | |
| 7.1 Prompts/cues | X | X | X | X | X | X | 6 |
| 7.2 Cue signaling reward | | X | X | X | | | 3 |
| 7.3 Reduce prompts/cues | | | X | | | | 1 |
| 12.1 Restructuring the physical environment | X | X | X | X | X | X | 6 |
| 12.2 Restructuring the social environment | | X | | | | | 1 |
| 12.5 Adding objects to the environment | X | X | X | X | X | X | 6 |
| Total | 3 | 5 | 5 | 4 | 3 | 3 | |
| **COERSION: creating expectation of cost \*** | | | | | | | |
| 1.6 Discrepancy between current behavior and goal | X | X | X | X | X | X | 6 |
| 1.8 Behavioral contract | X | X | | X | X | X | 5 |
| 1.9 Commitment | X | X | X | | X | X | 5 |
| 2.2 Feedback on behavior | X | X | X | X | X | X | 6 |
| 2.3 Self-monitoring of behavior | X | X | X | X | X | X | 6 |
| 2.4 Self-monitoring of outcome(s) of behavior | | X | X | X | | | 3 |
| 2.5 Monitoring of outcome(s) of behavior without feedback | | | X | | | | 1 |
| 2.6 Biofeedback | X | X | X | X | X | X | 6 |
| 2.7 Feedback on outcome(s) of behavior | | X | | X | | | 2 |
| 5.5 Anticipated regret | | X | | | | | 1 |
| 10.11 Future punishment | X | | | X | | | 2 |
| 14.3 Remove reward | | X | X | X | | | 3 |
| Total | 7 | 10 | 8 | 9 | 6 | 6 | |
| **INCENTIVIZATION: creating expectation of reward \*** | | | | | | | |
| 1.6 Discrepancy between current behavior and goal | X | X | X | X | X | X | 6 |
| 1.8 Behavioral contract | X | X | | X | X | X | 5 |
| 1.9 Commitment | X | X | X | | X | X | 5 |
| 2.2 Feedback on behavior | X | X | X | X | X | X | 6 |
| 2.3 Self-monitoring of behavior | X | X | X | X | X | X | 6 |
| 2.4 Self-monitoring of outcome(s) of behavior | | X | X | X | | | 3 |
| 2.5 Monitoring of outcome(s) of behavior without feedback | | | X | | | | 1 |
| 2.6 Biofeedback | X | X | X | X | X | X | 6 |
| 2.7 Feedback on outcome(s) of behavior | | X | | X | | | 2 |
| 7.2 Cue signaling reward | | X | X | X | | | 3 |
| 10.3 Non-specific reward | X | X | X | X | X | X | 6 |
| 10.4 Social reward | | X | | X | | | 2 |
| 10.5 Social incentive | | | | | X | | 1 |
| 10.6 Non-specific incentive | X | X | X | | | X | 4 |
| 10.8 Incentive (outcome) | X | X | | | | | 2 |
| 10.9 Self-reward | | | | | X | | 1 |
| 10.10 Reward (outcome) | | X | X | | | | 2 |
| 14.4 Reward approximation | | X | X | | | | 2 |
| 14.5 Rewarding completion | | X | X | X | | X | 4 |
| 14.6 Situation-specific reward | | X | | | | | 1 |
| 14.8 Reward alternative behavior | X | | | | | | 1 |
| 14.9 Reduce reward frequency | | X | | | | | 1 |
| Total | 10 | 18 | 13 | 11 | 9 | 9 | |

**Table 4.** EAM coding based on behavior change techniques (BCTs) associated with intervention functions of the Behaviour Change Wheel (BCW).

| Behavior Change Technique | Apple Watch (Series 4) | Fitbit Devices | Garmin Devices | Amazfit Bip | Galaxy Watch Active | Withings Steel HR | Total |
|---|---|---|---|---|---|---|---|
| **MODELING: providing an example \*** | | | | | | | |
| 6.1 Demonstration of the behavior | | X | | | | | 1 |
| Total | | 1 | | | | | |
| **PERSUASION: using communication to induce feelings of stimulate action \*** | | | | | | | |
| 2.2 Feedback on behavior | X | X | X | X | X | X | 6 |
| 2.6 Biofeedback | X | X | X | X | X | X | 6 |
| 2.7 Feedback on outcome(s) of behavior | | X | | X | | | 2 |
| 5.1 Information about health consequences | | X | X | | | | 2 |
| 5.6 Information about emotional consequences | | X | | | | | 1 |
| 6.2 Social comparison | X | X | X | X | X | X | 6 |
| 6.3 Information about others' approval | | X | | | | | 1 |
| 9.1 Credible source | | X | | | | | 1 |
| 13.2 Framing/reframing | | | X | | | | 1 |
| 15.3 Focus on past success | X | X | X | X | X | | 5 |
| Total | 4 | 9 | 6 | 5 | 4 | 3 | |
| **TRAINING: imparting skills \*** | | | | | | | |
| 2.2 Feedback on behavior | X | X | X | X | X | X | 6 |
| 2.3 Self-monitoring of behavior | X | X | X | X | X | X | 6 |
| 2.4 Self-monitoring of outcome(s) of behavior | | X | X | X | | | 3 |
| 2.6 Biofeedback | X | X | X | X | X | X | 6 |
| 2.7 Feedback on outcome(s) of behavior | | X | | X | | | 2 |
| 4.1 Instruction on how to perform the behavior | | X | X | | X | X | 4 |
| 4.4 Behavioral experiments | | | X | | | | 1 |
| 6.1 Demonstration of the behavior | | X | | | | | 1 |
| 8.1 Behavioral practice/rehearsal | | - | | | X | | 1 |
| 8.3 Habit formation | | X | X | X | X | X | 5 |
| 8.4 Habit reversal | | | X | | | | 1 |
| 8.7 Graded tasks | X | X | X | | | | 3 |
| 10.9 Self-reward | | | | | X | | 1 |
| 15.2 Mental rehearsal of successful performance behavior | | | X | | | | 1 |
| Total | 4 | 9 | 10 | 6 | 7 | 5 | |

\* intervention function definitions derived from Michie et al. [18].

The utility features of each device are presented in Table 5. The EAMs shared several of the same features: paired with a phone/tablet, synced with phone/tablet notifications, wrist worn, and tracked exercise sessions. In addition to PA, most EAMs tracked related health behaviors including sleep, nutrition, and sedentary behavior. Nutrition monitoring was available within the associated app or through a partnering external app (e.g., MyFitnessPal). The main difference between the "sedentary behavior" and "sitting time (alerts)" features was that sedentary behavior was overall idle time whereas sitting time (alerts) notified the wearer of prolonged periods of sitting. Available features were relatively common across all EAM devices with a few exceptions. Some devices only monitored minutes and energy expenditure from exercise, whereas other EAMs also monitored overall activity minutes and energy expenditure. The principal distinction between devices was the battery life which ranged from 1–2 days to more than 7 days.

**Table 5.** Utility features of electronic activity monitors.

| Features | Apple Watch | Fitbit Versa 2 | Fitbit Charge 3 | Fitbit Ionic (Adidas) | Garmin Vivomove HR | Garmin Vivosmart 4 | Amazfit Bip | Galaxy Watch Active | Witings Steel HR |
|---|---|---|---|---|---|---|---|---|---|
| **Functionality** | | | | | | | | | |
| Battery lasts 1–2 days | | | | | | | | X | |
| Battery lasts 3–4 days | X | | X | | X | X | | X | |
| Battery lasts 5–6 days | | X | | X | X | | | | |
| Battery last ≥ 7 days | | X | X | X | | | X | | X |
| Device pairs with a phone/tablet | X | X | X | X | X | X | X | X | X |
| Device syncs with phone/table notifications | X | X | X | X | X | X | X | X | X |
| Device face activity display | X | X | X | X | X | X | X | X | X |
| Mobile app activity display | X | X | X | X | X | X | X | | X |
| Wrist worn | X | X | X | X | X | X | X | X | X |
| **Behavior monitoring** | | | | | | | | | |
| Sleep | | X | X | X | X | X | X | X | X |
| Nutrition | | X | X | X | | | X | X | X |
| Sedentary | X | X | X | X | X | X | | X | X |
| Exercise (workout tracking) | X | X | X | X | X | X | X | X | X |
| **Physical activity tracking and other biometrics** | | | | | | | | | |
| Blood Pressure | | | | | | | | | X |
| Distance | X | X | X | X | | | | | |
| Energy expenditure (total) | X | X | X | X | X | X | X | | X |
| Exercise energy expenditure | X | X | X | X | X | X | X | X | X |
| Exercise minutes per day | X | X | X | X | X | X | X | X | X |
| Floors | | X | X | X | X | X | | | |
| GPS (built-in) | | | | | X | X | X | | |
| Heart rate | X | X | X | X | X | X | X | X | X |
| Hydration | | | | | X | X | | | X |
| Menstrual tracking/Female health | | X | X | X | X | X | | | X |
| Minutes per day (total) | | X | X | | X | X | X | | X |
| Sitting time (alerts) | X | X | X | X | | | X | X | |
| Steps per Day | X | X | X | X | X | X | X | X | X |
| Stress | | | | | X | | | X | |
| Water resistant | X | X | X | X | | X | | X | X |

## 4. Discussion

The purpose of the current study was to complete an updated content analysis of EAMs and provide context by aligning the results with components of the BCW model. In addition, the current study presented a review of utility features present among devices. On average, current EAMs implemented 27.7 BCTs and several features were similar across devices. Our results suggest that some EAMs are better equipped to address specific intervention functions than others, which suggests that organizing EAMs by their intervention functions may be a useful step in planning programs utilizing these devices.

The results of this study illustrate how EAMs have advanced in recent years [12,15–17]. Only nine EAMs were included in the current review, which is comparable to the 6–7 included in other analyses [15,16] but considerably less than the 13 devices included in the original publication [12]. This is partially due to a shrinking market of wearable manufacturers [25]. Functionally, all current devices were wrist worn whereas more wear locations were previously available, including the upper

arm and the lapel. The average BCTs implemented across the EAMs have increased compared to the average 14 to 26 in the previous reviews [12,15–17]. This increase in average BCTs may be the result of previous reviews using the CALO-RE taxonomy for coding [12,15,17]. The CALO-RE outlines 40 BCTs that are significantly correlated to PA [11] while the current study utilized the complete 93-item behavior change taxonomy [14] in order to translate the results to the BCW. In the present review, devices with the most implemented BCTs were Fitbit models (43 BCTs), followed by Garmin models (36 BCTs). Previously, Jawbone [12,17] and Withings [15] devices were found to implement the most BCTs. This further reflects the changes within the EAM industry. Jawbone no longer manufactures EAM devices, and Withings have expanded their focus to manufacture other health devices (e.g., sphygmomanometer, thermometer), while Fitbit has persisted and increased the number of BCTs implemented in their devices. Biofeedback, social support, social comparison, prompt/cues, and non-specific reward BCTs now appear to be standard across EAMs. Previously, these BCTs were seldomly implemented in devices. We also found an increase in behavioral contract and habit formation BCTs. Behavioral contract was coded as present if the user had to actively agree to a step or PA goal. Habit formation was related to an increase in sitting time alerts. EAMs often provided the idle alerts at regular intervals of inactivity and would instruct the wearer to stand, take a walk (e.g., take a few steps to meet the hourly goal), or otherwise move their body. The nature of this alert appeared intended to help form a moving habit.

### 4.1. Study Design Implications

Researchers and health practitioners can use these results to identify an EAM that best fits their intervention needs. This systematic review of features can help researchers and practitioners select an EAM that may increase the wearer's engagement with the device. Utility features such as battery life, wear location, and how the information is displayed can impact how the wearer will interact with the device [20,21,26]. Our results illustrate the different utility features among EAMs from the same manufacturer. Fitbit and Garmin devices have varying battery life depending on the model and the extent of use. For example, the Fitbit Charge 3 can last up to 7 days before charging or only 3–4 days if all features are enabled. Based on the features, some devices are better equipped for certain physical activities. The Fitbit Ionic (Adidas) has built-in GPS, which is a helpful feature for individuals who run outdoors, whereas the Fitbit Versa 2 and Charge 3 track total physical activity minutes, which is helpful for individuals who regularly perform non-leisure time physical activity. The Garmin Vivosmart 4 also has built-in GPS while the Garmin Vivomove HR does not. In addition, the Vivosmart 4 is water resistant, which makes it optimal for water-based exercise. If possible, researchers and practitioners should survey wearers prior to device selection to determine which features are most important to their research population. Once those data are collected, researchers can make an informed decision to select an EAM.

Furthermore, these results distinguish EAMs that support specific intervention function(s). This distinction is a critical step in designing interventions using the BCW [19]. The BCW consists of three layers: an inner layer that is derived from the COM-B model that identifies a source behavior; a middle layer that outlines a corresponding intervention function; and an outer layer that identifies policy [18,19]. In the context of this review, the source behavior is PA with the theoretical constructs of PA capability, motivation, or opportunity. Once the source behavior is identified, the appropriate intervention function and potential EAM device can be selected. Researchers can then pair the EAM and intervention function with policy. We suggest that researchers follow the BCW or a similar method for designing interventions and that they pay close attention to how BCTs in the EAM systems match their targeted theoretical constructs [19].

How exactly can researchers and health practitioners use these results? They must first identify an appropriate intervention function, preferably using the BCW [18]. Once they identify the intervention function, they can use Table 4 to identify an EAM that implements several BCTs for the given category. They can then use Table 5 to further compare devices and identify the EAM that best matches the

participant's needs. For example, if a researcher needed an EAM to intervene on PA as part of a large-scale employee wellness program and a needs assessment found that motivation was a major barrier, the researchers could review EAMs related to the persuasion intervention function. From there, the researchers may decide that a device with a long battery-life that also tracks lifestyle physical activity would be optimal for the wellness program. The researcher may then decide that the Fitbit Versa 2 fits their needs for the wellness program. Our results cannot be used to select an EAM that will guarantee an increase in individual's physical activity; rather the results offer a guide to align an EAM with user needs.

### 4.2. Strengths and Limitations

There are limitations to this study. First, the review is limited to the best-selling EAMs on CNet and it is not a comprehensive evaluation of all EAMs available on the market. This introduces some possible selection bias. However, this source provides a reliable list of EAMs and has been used as the primary source in previous evaluations [12,16]. Second, this review only evaluated the free version of the EAM and the associated app. It is possible that there were more implemented BCTs. The inter-coder reliability was slight to fair for some EAMs, this speaks to the adaptable nature of EAMS in that some features may also not be present. Three coders were used to identify the most BCTs to overcome this weakness. Additionally, coding was incomplete for the Galaxy watch and Apple watch due to app compatibility issues.

The current analysis focused on identifying the available features of the EAMs and the authors cannot determine which features will lead to increased engagement for the user without an intervention. Furthermore, our analysis does not account for the digital literacy of the user. EAM users should be educated on how to use and operate the device to take advantage of the features. Lastly, this analysis, similar to all EAM analyses, is limited because it cannot keep up with the rapid evolution of the devices. However, the process on how to select an EAM to be integrated into an intervention remains the same despite rapid changes in the devices.

There were many strengths of this investigation. Different models from the same manufacturer were reviewed. Although the behavioral content analysis was the same between these devices, the practical features differed, which provides further considerations for intervention design. To our knowledge, this review includes the first systematic evaluation of device features that may impact engagement and overall wearer adherence. The biggest strength of this study is that the BCT coding was presented in relation to the intervention functions of the BCW. This presentation allows researchers to select an EAM that best fits the context and targets of their planned intervention.

## 5. Conclusions

This study aimed to perform an updated behavioral content analysis of EAMs that evaluated utility features and aligned the results with the BCW. EAMs included in this review were Apple Watch Nike Series 5, Fitbit Versa 2, Fitbit Charge 3, Fitbit Ionic—Adidas Edition, Garmin Vivomove HR, Garmin Vivosmart 4, Amazfit Bip, Galaxy Watch Active, and Withings Steel HR. The devices shared several of the same utility features while battery life varied. The devices also shared several of the same BCTs, but Fitbit devices implemented the most BCTs that support the majority of the BCW intervention functions. Researchers and health practitioners can use these results to select appropriate EAMs for their intervention needs.

**Supplementary Materials:** The following are available online at http://www.mdpi.com/2227-7080/8/4/75/s1, Additional File S1: Complete coding table; Additional File S2: Complete coding sheets; Additional File S3: Behavior change technique examples in EAM.

**Author Contributions:** Z.H.L. designed and managed the study, coded EAMs, and wrote the manuscript. M.C. and G.R. coded EAMs. M.C.S. and E.J.L. assisted in the analysis strategy. All authors have read and agreed to the published version of the manuscript.

**Funding:** This research was funded by a Research, Scholarship, and Creative Activity minigrant from California State Polytechnic University, Pomona. The funding source had no role in the study design, data collection, management, analysis, interpretation, or preparation of the manuscript.

**Conflicts of Interest:** The authors declare no conflict of interest.

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
