# Peer review of "Analysis of the Behavioral Change and Utility Features of Electronic Activity Monitors"

_technologies, doi:10.3390/technologies8040075_

Round 1
Reviewer 1 Report
The paper is well written and clear about its objectives.
The study proposes to complete an updated content analysis of Electronic Activity Monitors and align the results with components of the Behaviour Change Wheel model that impact intervention design and, potentially, wearer adherence. At this step an illustration of EAM and BCW would be useful to well understand the concepts.
The question of the impact on general user engagement and adherence from "quantified self" connected objects is a major issue. The relation between behavior change techniques and the intervention functions of the Behaviour Change Wheel is very interesting.
However, the scientific impact is not so obvious. How can we be sure that this study is relevant in practice and for what kind of users?
Who are these EAMs aimed at? Whether it is young people, technology enthusiasts, or older people who are in a digital divide or have a poor knowledge of technology, it is not the same thing. This should be a discussion in a specific section.
The adoption of these tools to assist physical exercise practices is linked to the ease of use of watches but also to their energy autonomy and accessibility, this aspect seems to have been overlooked.
What is the profile of the trained coders recruited? Young people who may or may not be familiar with these technologies? How did the coding for each function realize? Was it only yes/no based on a questionnaire? It would have been interesting to have these different EAMs tested by a sample of people and to have their real feedback on the BCTs proposed by the different devices.
This revision/testing work should be reinforced by a more in-depth analysis of field data on the people tested in real situations in order to assess the relevance of the proposed tool.
What means SMD on the introduction section?
Author Response
Reviewer 1
The paper is well written and clear about its objectives.
Comment 1: The study proposes to complete an updated content analysis of Electronic Activity Monitors and align the results with components of the Behaviour Change Wheel model that impact intervention design and, potentially, wearer adherence. At this step an illustration of EAM and BCW would be useful to well understand the concepts.
Response 1: We agree that the manuscript did not provide enough detail on the behavioral aspect of the study. Within the introduction (lines 52-56), we provided more details on how the behavior change techniques were identified in a taxonomy. Also within the introduction, we provided examples of how the intervention functions within the Behaviour Change Wheel can lead to a behavior change technique (lines 79-87). We also provided example illustrations of the EAM and behavior change techniques (additional file 3). For copyright reasons we could not include an illustration of the BCW.
Comment 2: The question of the impact on general user engagement and adherence from "quantified self" connected objects is a major issue. The relation between behavior change techniques and the intervention functions of the Behaviour Change Wheel is very interesting.
Response 2: We cannot determine ultimate impact on user engagements because we did not enroll study participants to use the EAMs for behavior change. The research suggests that the features we identified should positively impact user engagement. Throughout the manuscript we changed the language to clarify that our study does not directly evaluate user engagement. Specifically, we added “Our results cannot be used to select an EAM that will definitively increase individual physical activity; rather the results offer a guide to align an EAM with user needs (lines 252-254).”
Comment 3: However, the scientific impact is not so obvious. How can we be sure that this study is relevant in practice and for what kind of users?
Response 3: We believe that this study is relevant to anyone that uses an EAM for physical activity promotion. We revised our study aim section to clarify the purpose of the study: “Our analysis was expanded to include a systematic review of device utility features and the results are aligned with intervention function within the BCW. This was done to help align EAM features with interventions needs. Researchers and health practitioners can use the results to make an informed selection of an EAM for physical activity promotion (lines 100-103).” We also added a section under the Study design implications that outlines exactly how researchers and health practitioners can use these results to improve their practice (lines 243-254).
Comment 4: Who are these EAMs aimed at? Whether it is young people, technology enthusiasts, or older people who are in a digital divide or have a poor knowledge of technology, it is not the same thing. This should be a discussion in a specific section.
Response 4: Our study was aimed at researchers and health practitioners who plan to use the EAMs with participants/patients. The target population can vary widely and we agree that certain subgroups may have challenges with these devices. We added the following statement to our study limitations “Furthermore, our analysis does not account for the digital literacy of the user. EAM users should be educated on how to use and operate the device to take advantage of the features (lines 267-268).”
Comment 5: The adoption of these tools to assist physical exercise practices is linked to the ease of use of watches but also to their energy autonomy and accessibility, this aspect seems to have been overlooked.
Response 5: We appreciate that the reviewer acknowledges the potential of EAMs. We added the following statement to the introduction: EAMs also allow for the wearer’s behavior to be more accessible to them through self-monitoring and allow for real-time feedback, which may promote autonomy. (lines 44-46).
Comment 6: What is the profile of the trained coders recruited? Young people who may or may not be familiar with these technologies? How did the coding for each function realize? Was it only yes/no based on a questionnaire? It would have been interesting to have these different EAMs tested by a sample of people and to have their real feedback on the BCTs proposed by the different devices.
Response 6: We agree that it would be interesting to gather feedback on the BCT among a sample of EAM users. It was beyond the scope of this study but an interesting area for future research. We expanded on the training and qualifications of the coders within the text (lines 118-123).
Comment 7: This revision/testing work should be reinforced by a more in-depth analysis of field data on the people tested in real situations in order to assess the relevance of the proposed tool.
Response 7: As mentioned in a previous comment, we agree that this would be an interesting investigation, but it is beyond the scope of this study. The goal of the current study was to present the information so that researchers and health practitioners can select a device that better aligns with their needs. The next logical research step is to investigate whether user engagement and physical activity outcomes are improved after using such a tool. This type of investigation is currently being proposed by the PI.
Comment 8: What means SMD on the introduction section?
Response 8: SMD stands for Standardized Mean Difference. We wrote this out in text (lines 47-48).
Reviewer 2 Report
The article presents a comparison of several Electronic Activity Monitors in which different features are analyzed in terms of potential user adherence and impact in behavioral therapy. Although the content of the article may be of interest to the audience of the journal, the paper presents the following issues:
- the sample is defined based on one source only (CNet), whereas it would have been more appropriate to select it according to the ranking of multiple outlets, to avoid a potentially biased list of participants
- although the study consists of a heuristic analysis, it is missing a reference framework. The paper refers to "trained coders", but it does not describe the method utilized for training them, or the model according to which coders have been trained. Also, the coders are the authors of the paper, which introduces additional bias
- the results are not confirmed or compared to any previous study or reference, which impacts the applicability of the article
As a side note, there are several verbose sentences that might make it hard for the reader to digest the article. Dividing them into smaller chunks or simplifying them would improve the overall readability.
Author Response
Reviewer 2
The article presents a comparison of several Electronic Activity Monitors in which different features are analyzed in terms of potential user adherence and impact in behavioral therapy. Although the content of the article may be of interest to the audience of the journal, the paper presents the following issues:
Comment 1: the sample is defined based on one source only (CNet), whereas it would have been more appropriate to select it according to the ranking of multiple outlets, to avoid a potentially biased list of participants
Response 1: We agree that using one source may have introduced some selection bias to our EAM sample. This is described in the limitation section “First, the review is limited to the best-selling EAMs on CNet and it is not a comprehensive evaluation of all EAMs available on the market. This introduces some possible selection bias (lines 255-257).” However, CNet is a commonly used as the main source for this type of research. We added the following to explain this point “However, this source provides a reliable list of EAMs and has been used as the primary source in previous evaluations [12,16] (lines 257-258).”
Comment 2: although the study consists of a heuristic analysis, it is missing a reference framework.
Response 2: There is not a specific framework for our analysis. Rather, analysis was based on whether established behavior change techniques and utility features were present or not. We followed the same analytic approach used in previous EAM content analyses. Citations include:
- Lyons, E.J.; Lewis, Z.H.; Mayrsohn, B.G.; Rowland, J.L. Behavior change techniques implemented in electronic lifestyle activity monitors: a systematic content analysis. Journal of medical Internet research 2014, 16, e192.
- Mercer, K.; Li, M.; Giangregorio, L.; Burns, C.; Grindrod, K. Behavior change techniques present in wearable activity trackers: a critical analysis. JMIR mHealth and uHealth 2016, 4, e40.
- Chia, G.L.C.; Anderson, A.; McLean, L.A. Behavior change techniques incorporated in fitness trackers: content analysis. JMIR mHealth and uHealth 2019, 7, e12768.
- Duncan, M.; Murawski, B.; Short, C.E.; Rebar, A.L.; Schoeppe, S.; Alley, S.; Vandelanotte, C.; Kirwan, M. Activity trackers implement different behavior change techniques for activity, sleep, and sedentary behaviors. Interactive journal of medical research 2017, 6, e13.
Comment 3: The paper refers to "trained coders", but it does not describe the method utilized for training them, or the model according to which coders have been trained. Also, the coders are the authors of the paper, which introduces additional bias
Response 3: We agree that more detail about the coders was missing in the original manuscript. We expanded on the training and qualifications of the coders within the text (lines 118-123). The coders performed the analysis for this study which is a reasonable contribution for authorship.
Comment 4: the results are not confirmed or compared to any previous study or reference, which impacts the applicability of the article
Response 4: We elected to present the confirmation and comparison of the results in the Discussion section of the manuscript. We compare our results with the results of four similar studies (lines 192-214).
Comment 5: As a side note, there are several verbose sentences that might make it hard for the reader to digest the article. Dividing them into smaller chunks or simplifying them would improve the overall readability.
Response 5: Thank you for bringing this to our attention. We simplified the text throughout the manuscript.
Reviewer 3 Report
Comments to the Author
The authors studied the characteristics of electronic activity monitors and the monitor behavior change techniques. The use of activity monitors is widely extended and this is a very interesting topic. I have some recommendations to the authors on how to strengthen their work.
1.The title should be more informative, “Analysis of the behavioral change features of EAM” or something similar.
2. The Introduction section could provide a clear and strong justification for the clear purpose of the study. I recommend changing lines 57-66.
3. The Methods section should be ordered in parts. I think that authors should implement a scoring protocol provide results according to this protocol instead the current descriptive analysis. A descriptive analysis as presented is not enough in order to provide scientific evidence in a scientific journal. This is my main concern, and I think that it is a serious one.
4. Figure 1 could be changed. Instead the current one, authors should compare each device (bars representing each change technique and grouped for each device)
5. Table 3 could be attached as supplementary file. Again, authors should establish a scoring protocol.
6. Lines 219-220: I do not think that this could be a limit. Instead this, I think that the rapid evolution of EAM could be an important one.
Author Response
Reviewer 3
The authors studied the characteristics of electronic activity monitors and the monitor behavior change techniques. The use of activity monitors is widely extended and this is a very interesting topic. I have some recommendations to the authors on how to strengthen their work.
Comment 1: The title should be more informative, “Analysis of the behavioral change features of EAM” or something similar.
Response 1: We appreciate the suggestion. We revised the title to “Analysis of the Behavioral Change and Utility Features of Electronic Activity Monitors.”
Comment 2: The Introduction section could provide a clear and strong justification for the clear purpose of the study. I recommend changing lines 57-66.
Response 2: Thanks to the recommendations from all of the reviewers, we revised our introduction. We believe that the revisions and additions provide a more streamlined introduction (lines 31-102). We also clarified our study aim which now reads “The aim of the current study was to perform an updated behavioral content analysis of EAMs currently on the market. Our analysis was expanded to include a systematic review of device utility features and the results are aligned with intervention function within the BCW. This was done to help align EAM features with interventions needs. Researchers and health practitioners can use the results to make an informed selection of an EAM for physical activity promotion. (lines 98-102).”
Comment 3: The Methods section should be ordered in parts. I think that authors should implement a scoring protocol provide results according to this protocol instead the current descriptive analysis. A descriptive analysis as presented is not enough in order to provide scientific evidence in a scientific journal. This is my main concern, and I think that it is a serious one.
Response 3: We recognize that our analysis lacks a scoring protocol that ranks the EAMs based on present features. We also recognize that the needs of each researcher and health practitioner is different and some EAMS may be more appropriate based on their needs. For this reason, our analysis is descriptive. We added a section in the discussion to elaborate how the scientific evidence presented can be used within interventions (lines 242-253).
Comment 4: Figure 1 could be changed. Instead the current one, authors should compare each device (bars representing each change technique and grouped for each device)
Response 4: We modified Figure 1 based on the reviewer’s recommendations. The figure still illustrates the frequency of behavior change techniques by intervention function, but it directly compares the frequency across EAMs.
Comment 5: Table 3 could be attached as supplementary file. Again, authors should establish a scoring protocol.
Response 5: Our analysis included behavior change techniques as well as the utility features. For this reason, we believe that Table 3 should be included in the text instead of an additional file. We included further discussion based on these results. Our discussion now describes the possible implications of utility features (lines 220-228) as well as how this table can be used to select an appropriate EAM (lines 242-253).
Comment 6: Lines 219-220: I do not think that this could be a limit. Instead this, I think that the rapid evolution of EAM could be an important one.
Response 6: Based on the feedback from another reviewer, we kept the limitation of using CNet as a source. We also added this recommended limitation “Lastly, this analysis, like all EAM analyses, is limited because it cannot keep up with the rapid evolution of the devices (lines 268-269).”
Round 2
Reviewer 1 Report
Thank you to the authors for making a great effort to answer to each request from Reviewer 1.
The additions are relevant and the illustration added makes the discussion more concrete.
The article is clearer on its objectives even if its scientific scope is somewhat limited because it is limited to offer a guide to align an EAM with user needs and does not include an initial deployment on few people, which would have made it possible to obtain initial feedback from the users (researcher, health practitioner, patient) and to assess its performance or practical usefulness. However, this corrected version makes this paper more soundness.
Reviewer 2 Report
The authors addressed the main concerns of the reviewer.
Reviewer 3 Report
Authors have improved their manuscript
regards